# Detection of Germline Variants in 450 Breast/Ovarian Cancer Families with a Multi-Gene Panel Including Coding and Regulatory Regions

**DOI:** 10.3390/ijms22147693

**Published:** 2021-07-19

**Authors:** Chiara Guglielmi, Rosa Scarpitta, Gaetana Gambino, Eleonora Conti, Francesca Bellè, Mariella Tancredi, Tiziana Cervelli, Elisabetta Falaschi, Cinzia Cosini, Paolo Aretini, Caterina Congregati, Marco Marino, Margherita Patruno, Brunella Pilato, Francesca Spina, Luisa Balestrino, Elena Tenedini, Ileana Carnevali, Laura Cortesi, Enrico Tagliafico, Maria Grazia Tibiletti, Stefania Tommasi, Matteo Ghilli, Caterina Vivanet, Alvaro Galli, Maria Adelaide Caligo

**Affiliations:** 1SOD Molecular Genetics, University Hospital of Pisa, 56126 Pisa, Italy; chiara.guglielmi.cg@gmail.com (C.G.); eleonoraconti4@gmail.com (E.C.); mariella.tancredi@gmail.com (M.T.); cunegonda_f@yahoo.it (E.F.); c.cosini@ao-pisa.toscana.it (C.C.); 2Division of Pathology, University of Pisa, 56126 Pisa, Italy; rose2312@hotmail.it; 3Department of Clinical and Experimental Medicine, University of Pisa, 56126 Pisa, Italy; gaetana.gambino@gmail.com; 4Functional Genetics and Genomics Laboratory, Institute of Clinical Physiology, IFC-CNR, 56127 Pisa, Italy; f.belle01@hotmail.com (F.B.); tizicerv@ifc.cnr.it (T.C.); 5Section of Oncological Genomics, Fondazione Pisana per la Scienza, 56017 Pisa, Italy; p.aretini@fpscience.it; 6Division of Internal Medicine, University Hospital of Pisa, 56126 Pisa, Italy; c.congregati@ao-pisa.toscana.it; 7Department of Life Sciences, University of Modena and Reggio Emilia, 41125 Modena, Italy; marcomarino83@gmail.com (M.M.); elena.tenedini@unimore.it (E.T.); enrico.tagliafico@unimore.it (E.T.); 8IRCCS Istituto Tumori “Giovanni Paolo II”, 70124 Bari, Italy; m.patruno@oncologico.bari.it (M.P.); brunellapilato@gmail.com (B.P.); s.tommasi@oncologico.bari.it (S.T.); 9SC Medical Genetics, ASSL Cagliari, 09126 Cagliari, Italy; francescaspina@hotmail.com (F.S.); luibale@hotmail.com (L.B.); cateviva@libero.it (C.V.); 10Ospedale di Circolo ASST Settelaghi, 21100 Varese, Italy; ileana.carnevali@asst-settelaghi.it (I.C.); mariagrazia.tibiletti@asst-settelaghi.it (M.G.T.); 11Department of Oncology, Haematology and Respiratory Diseases, University Hospital of Modena, 41124 Modena, Italy; cortesilaura67@gmail.com; 12Breast Cancer Center, University Hospital, 56126 Pisa, Italy; m.ghilli@ao-pisa.toscana.it

**Keywords:** hereditary breast and ovarian cancer, BRCA1/2, breast cancer predisposition genes, coding variants, non-coding variants, regulatory regions, gene panel, NGS

## Abstract

With the progress of sequencing technologies, an ever-increasing number of variants of unknown functional and clinical significance (VUS) have been identified in both coding and non-coding regions of the main Breast Cancer (BC) predisposition genes. The aim of this study is to identify a mutational profile of coding and intron-exon junction regions of 12 moderate penetrance genes (*ATM*, *BRIP1*, *CDH1*, *CHEK2*, *NBN*, *PALB2*, *PTEN*, *RAD50*, *RAD51C*, *RAD51D*, *STK11*, *TP53*) in a cohort of 450 Italian patients with Hereditary Breast/Ovarian Cancer Syndrome, *wild type* for germline mutation in *BRCA1/2* genes. The analysis was extended to 5′UTR and 3′UTR of all the genes listed above and to the BRCA1 and BRCA2 known regulatory regions in a subset of 120 patients. The screening was performed through NGS target resequencing on the Illumina platform MiSeq. 8.7% of the patients analyzed is carriers of class 5/4 coding variants in the *ATM* (3.6%), *BRIP1* (1.6%), *CHEK2* (1.8%), *PALB2* (0.7%), *RAD51C* (0.4%), *RAD51D* (0.4%), and *TP53* (0.2%) genes, while variants of uncertain pathological significance (VUSs)/class 3 were identified in 9.1% of the samples. In intron-exon junctions and in regulatory regions, variants were detected respectively in 5.1% and in 32.5% of the cases analyzed. The average age of disease onset of 44.4 in non-coding variant carriers is absolutely similar to the average age of disease onset in coding variant carriers for each proband’s group with the same cancer type. Furthermore, there is not a statistically significant difference in the proportion of cases with a tumor onset under age of 40 between the two groups, but the presence of multiple non-coding variants in the same patient may affect the aggressiveness of the tumor and it is worth underlining that 25% of patients with an aggressive tumor are carriers of a *PTEN* 3′UTR-variant. This data provides initial information on how important it might be to extend mutational screening to the regulatory regions in clinical practice.

## 1. Introduction

Breast cancer (BC) is the most common malignant tumor in women representing 25% of the total number of cancer cases in women with an annual increase of 793,700 new cases and 197,600 deaths only in developed countries [1,2].

The presence of germline mutations in the *BRCA1* and *BRCA2* genes significantly increases the risk of developing breast and ovarian cancer [3,4,5], however only 25% of the risk of familial cancer can be attributed to pathogenetic mutations localized in the coding regions of these two genes. Most of the risk is probably due to a combination of high, moderate, and low risk variants which may also include non-coding variants located in regulatory regions [4].

In clinical practice, during the past years, the mutational analysis in patients with Hereditary Breast and Ovarian Cancer (HBOC) has been mostly limited to the coding regions and to the intron-exon junctions of *BRCA1* and *BRCA2*, precluding the identification of mutations in the non-coding and/or regulatory regions and in other genes that can confer a high or moderate risk to the disease. Under these conditions, mutation screening is negative in 80% of the cases analyzed [6].

Segregation analysis generated no evidence for further dominant genes with a risk penetrance profile comparable to *BRCA1* or *BRCA2* and this has been corroborated by linkage studies [7,8]. Over the years, efforts have been made to identify new intermediate-penetrance predisposition genes, although it is difficult to predict how many they might be and what proportion of the total familial risk is attributable to them. The known intermediate-penetrance genes are all involved in DNA repair or have a role in BRCA1 or BRCA2 pathway, so these genes represent plausible candidates and are the most intensively investigated [9]. In addition, in recent years, the interest in investigating the effect that regulatory variants could have on cancer risk has increased [10].

Generally, gene expression is controlled at various levels by key regulatory elements located in promoters, introns, in “long-range enhancers” and in Untranslated Regions (5′and 3′ UTR). This is also true for *BRCA1* and *BRCA2*; in fact, it has been shown that the activity of the 5′UTR and of the *BRCA1/2* exon-flanking regions is controlled by a self-regulation mechanism and by an increasing number of transcription factors that act as activators or repressors. Interestingly, germline variants of 5′UTR causing a significant change in promoter activity have been identified in such regions [11,12,13].

Similarly, germline variants localized in consensus regions for “RNA binding protein” and miRNA located in 3′UTR of *BRCA1* [14,15] and *PTEN* [16,17] have been associated with breast cancer risk.

Therefore, germline regulatory variants can be associated with a deregulation of the transcriptional activity and modify the physiological expression both of BRCA genes and of other genes that can affect the risk of breast cancer [13].

With the progression of DNA sequencing technologies, it is possible to analyze in each patient multiple genomic regions by simultaneously screening multiple target genes.

An ever-increasing number of variants of unknown functional and clinical significance (VUS) have been identified in both coding and non-coding regions. Therefore, it is absolutely necessary to establish their risk level in carrier patients, for more effective genetic counselling.

In 2008, the International Agency for Research on Cancer (IARC) proposed to classify the missense variants on the basis of their probable pathogenicity, calculated by multifactorial models, in five risk classes: Class 1 (non-pathogenic or of no clinical significance), Class 2 (likely non-pathogenic or of little clinical significance), Class 3 (uncertain), Class 4 (likely pathogenic), and Class 5 (definitely pathogenic) [18].

Recently, the indications of the American College of Medical Genetics (ACMG) have been increasingly followed by several clinicians and research groups. They have recommended a five-tier classification system where variants can be classified as: ‘Pathogenic’, ‘Likely pathogenic’, ‘Uncertain significance’, ‘Likely benign’, and ‘Benign’ [19]. 

In this work we used this classification that, among those available, is the most complete and is based on an algorithm that takes into account data on penetrance, clinical information of carriers, and scientific evidence.

To classify VUSs, alternative methods have also been developed. They do not directly evaluate the role of the variant in the development of cancer, but the structural and functional modifications of the proteins of interest by providing indirect evidence of the pathogenicity of the variant [20,21]. Both in vitro functional assays and in silico predictors can be used [20].

Recently the international consortium ENIGMA (Evidence-based Network for the Interpretation of Germline Mutant Alleles) has become in charge of reviewing and comparing all the available and useful functional tests in order to define the functional meaning of a variant and to contribute to the interpretation of clinical data [22,23,24,25].

In this study involving five Italian hospital centers (Pisa, Cagliari, Varese, Modena, and Bari), a genomic screening was conducted on 450 Breast and/or Ovarian Cancer patients with cancer family history in at least two generations and wild type for germinal mutations in the coding regions and/or for large rearrangements in BRCA1/2 genes.

The analysis was carried out by Next Generation Sequencing (NGS) using a panel that covers the coding regions and the intron-exon junctions of 10 genes mainly involved in DNA repair pathways: *ATM*, *BRIP1*, *CDH1*, *CHECK2*, *PALB2*, *PTEN*, *RAD51C*, *RAD51D*, *STK11*, *TP53*. For 120 of these patients, the screening was extended to RAD50 and NBN genes and to 5′UTR and 3′UTR of all the genes listed above including the known regulatory regions of *BRCA1* and *BRCA2*. We hypothesized that variants located in the regulatory regions of the main BC predisposition genes can contribute to tumor development in a manner comparable to the variants located in coding regions. To support this hypothesis, we compared the average age of disease onset between the coding and the non-coding variant carriers with the same cancer type. Moreover, we analyzed the proportion of cases with a tumor onset under the age of 40 between the two groups and we verified any variant association in more aggressive tumours.

## 2. Results

### 2.1. Summary Overview of Germline Variants Found

Mutational screening was performed for 450 patients in the coding regions and 50 bp flanking intronic sequences of 10 genes: *ATM*, *BRIP1*, *CDH1*, *CHECK2*, *PALB2*, *PTEN*, *RAD51C*, *RAD51D*, *STK11*, *TP53*. For 120 of these patients, the analysis was extended to two additional genes, *RAD50* and *NBN*, and to the regulatory regions of *BRCA1* and *BRCA2*, 5′UTR, and 3′UTR for all 12 genes listed above. 

We identified 88 coding variants in 91 patients distributed as follows:-16 class 4 (Likely-pathogenic) and 19 class 5 (Pathogenic) variants in 39 patients (8.7%)-37 class 3 (VUS) in 41 patients (9.1%)-16 Likely-benign variants in 19 patients (4.2%), but all of them potentially classifiable as VUS, according to ACMG classification, because of interpretation conflicts.

In the intron-exon junctions, 13 variants in 23 patients were identified (5.1%).

The extension of the screening in 120 patients to the known regulatory regions of *BRCA1/2* and to 5′UTR, 3′UTR for all the genes analyzed, allowed us to identify another 51 variants in 39 patients (32.5%).

### 2.2. Coding Variants

A total of 88 coding variants were identified: 35 Pathogenic or Likely-pathogenic variants, 37 VUS, and 16 Likely-benign variants.

Thirty out of 35 Pathogenic or Likely-pathogenic identified coding variants were encoded in dbSNP and 27 were already known in literature or reported in databases such as IARC, LOVD, MGeND, ClinVar Miner with more clinical information. Five variants have never been registered: c.8368delA in *ATM*, c.507delT and c.1430delC in *CHEK2*, c.1565dupC in *PALB2*, c.397_398delCA in *RAD51C*. The 35 class 4 and 5 variants are distributed in the following genes: 14 in *ATM*, 5 in *BRIP1*, 8 in *CHEK2*, 3 in *PALB2*, 2 in *RAD51C*, 2 in *RAD51D*, and 1 in *TP53*.

Thirty-four out of 37 VUS identified were encoded in dbSNP and 25 of these were already known and reported in databases previously cited. The following 3 VUSs were completely new: c.2317A > G in ATM, c.3029A > G in BRIP1 and c.17_18insGCG in *PTEN*. Eighteen of the 37 VUSs identified are in *ATM*, 5 in *BRIP1*, 2 in *CDH1*, 1 in *CHEK2*, 3 in *NBN*, 3 in PALB2, 2 in *PTEN*, 2 *in RAD50*, and 1 in *STK11*.

Fifteen out of 16 Likely-benign variants found had a dbSNP entrie and 12 of these were encoded in databases. One variant had never been registered (c.252T > C in *RAD51D*). The 16 Likely-benign variants are distributed as follows: 4 in *ATM*, 1 in *BRIP1*, 3 in *CDH1*, 1 in *CHEK2*, 2 in *RAD50*, 1 in *RAD51C*, 3 in *RAD51D*, and 1 in *STK11*.

Figure 1 shows the percentage of distributions of the variants stratified by gene, according to the current ACMG classification. As shown in Figure 1
*ATM* is the gene with the highest mutation rate. 

Sixty-six out of 88 identified variants are missense variants (of which 18 Pathogenic and Likely-pathogenic, 35 VUS and 13 Likely-benign), 5 Nonsense variants (all Pathogenic), 4 Synonymous variants (1 VUS and 3 Likely-benign), 11 frameshift variants (all Pathogenic), and 2 in-frame variants (1VUS and 1 Pathogenic). The distribution by gene is shown in Figure 2.

The predictive analysis on the effect of amino acid change on the function of the protein was conducted with PROVEAN and Polyphen2 software, as described in Materials and Methods. Specifically, the information for each identified variant is shown in Appendix A, where it is also reported whether the variant has already been identified and registered in database by other groups.

### 2.3. Demographics and Clinical Features of Patiens with Coding Variants

The average age of tumor diagnosis for 91 patients with coding variants is 44. 8.8% (8) of the patients have a bilateral BC, 2.2% (2) have both breast and ovarian cancer, 4.4% (4) have ovarian cancer, 2.2% (2) have multiple tumors (in addition to breast cancers) and the remaining percentage, 82.4% (75) of total cases, has a BC. Among the latter patients, 16% (12) have Triple-Negative Breast Cancer (TNBC).

Their family histories are distributed as follows: 48.3% (44) have Hereditary Breast Cancer (HBC), 12.1% (11) have Hereditary Breast and Ovarian Cancer (HBOC), 18.7% (17) have Early Onset Breast Cancer (EOBC), and 20.9% (19) have other Hereditary Breast Cancer Syndromes (HBCS) (Table 1).

Thirty-nine patients are carriers of variants classified as “Pathogenic” or “Likely-pathogenic”, 41 patients are carriers of VUS, while “Likely-benign” variants have been identified in 19 patients. If we consider the total number of patients screened in the coding regions (450) the percentage of class 4 and 5 variant carriers is 8.7% and it is distributed in this way: 3.6% have variants in *ATM*, 1.6% in *BRIP1*, 1.8% in *CHEK2*, 0.7% in *PALB2*, 0.4% in *RAD51C*, 0.4% in *RAD51D*, and 0.2% in *TP53*. In Figure 3 the frequencies of patients with Pathogenic and Likely-pathogenic, VUS and Likely-benign variants are reported.

Among the 75 patients with BC, we identified 82 variants of which 30 Pathogenic and Likely-pathogenic, broken down as follows: 13 in *ATM* gene, 5 in *BRIP1*, 2 in *PALB2*, 6 in *CHEK2*, 2 in *RAD51C*, 1 in *RAD51D* and 1 in *TP53*. In these same patients we identified 38 VUS: 15 in *ATM*, 6 in *BRIP1*, 4 in *CDH1*, 3 in *NBN*, 3 in *PALB2*, 2 in *CHEK2*, 2 in *PTEN*, 1 in *RAD50*, 1 in *RAD51D* and 1 in *STK11*. The 14 remaining variants found are classified as Likely-benign and distributed as follows: 6 in *ATM*, 3 in *RAD51D*, 2 in *CDH1* and 3 respectively in *BRIP1*, *RAD50*, and *RAD51C* genes. 

Among the four cases of patients with ovarian cancer, we found a Pathogenic variant in *RAD51C*, 2 VUSs, respectively in *ATM* and *RAD50* genes, and a Likely-benign variant in *STK11*.

Among the 12 cases of multiple tumors in coding variant carriers (9 cases of bilateral breast cancer, 2 of breast and ovarian cancer, and 1 case of breast tumor associated with pancreatic cancer), 9 have Pathogenic or Likely-pathogenic variants, 1 VUS and 2 Likely-benign variants. The statistical analysis has found a significant association of these cases with Pathogenic variant carriers (χ2-test, *p* < 0.01) in comparison with the cases of single tumors.

Three out of nine patients with Pathogenic variants have variants in *ATM*, 4 in *CHEK2*, 1 in *BRIP1* and 1 in *PALB2* (Table 2). Three of these nine variants are new variants never reported in dbSNP. One case has a VUS in *ATM* gene and two patients with multiple tumors have Likely-benign variants in CDH1 and *CHEK2* genes. Frequency of coding variants in *CHEK2* gene is significant in these patients (Fisher test, *p* < 0.05).

By analyzing cases of more aggressive tumors such as the 12 Triple-negative breast cancer patients (TNBC) (Table 3), all with G2 or G3 infiltrating ductal carcinoma, we identified 6 Pathogenic and Likely-pathogenic variants: 1 in *ATM*, 1 in *BRIP1*, 2 in *CHEK2*, 1 in *RAD51D* and 1 in *TP53*. In 6 patients, 4 VUSs (3 in ATM and 1 in *PTEN* never recorded before) and 2 Likely-benign variants were identified. The latter all have a conserved evolutionary profile (GERP > 3.8) and a conflict of interpretation according to Varsome and ClinVar.

### 2.4. Non-Coding and/or Regulatory Variants

The screening of non-coding regions has allowed us to identify 64 variants. Forty-six of these have a dbSNP code and only 24 are known variants already registered in databases by other groups.

The 64 variants are distributed as follows: 18 are located at 5′UTR, 21 at 3′UTR, 6 in splice site regions, and 19 in intron regions.

Figure 4 shows the distribution of the variants by gene divided according to their location.

*ATM*, *BRCA1*, *BRCA2*, and *PTEN* are among the genes in which we have found more non-coding variants (Figure 4).

The variant putative effect on gene expression, was carried out by using different software (described in Materials and Methods) depending on variant position. The results are reported in Appendix A.

An unexpectedly high number of variants in 5′ UTR of *PTEN* (4) and *RAD51C* (3) was found and, according to our analysis, all of them are localized in TF binding regions (Appendix A).

The in-silico analysis performed by ‘TFbind’ and ‘Promo’ software, has identified TF consensus sequences in the regions where the 5′UTR of *BRCA1* (c.-676T > A) and *BRCA2* (c.-1193C > T) variants are localized (Appendix A).

In 3′UTR, the greatest contribution is due to variants located in *ATM*, *PTEN*, and *RAD50* genes (Figure 4) on which putative miRNA-binding regions have been identified by ‘TargetScanHuman’ and ‘miRDB’ software (Appendix A).

In splicing regions, we identified six variants and only one, c.904+5G > T, was predicted to inactivate the wild-type donor site in the intron 6 of the *RAD51C* gene, and therefore is classified by the ACMG criteria as ‘Pathogenic’.

Three variants were in *ATM* but only the c.7788+ 8G > T, localized in intron 52, was predicted by ‘Human Splicing Finder’ as a possible modifier of an exonic ESE site with a potential alteration of splicing. The remaining 2 in *CHEK2* and *BRIP1* genes were predicted neutral on splicing (Appendix A).

In the remaining intronic regions, variants in *ATM*, *BRCA1*, *BRCA2*, *CHEK2*, *NBN*, *RAD50*, *STK11*, and *TP53* genes were identified. The possible presence of enhancer regions, compatible with bonds to transcription factors, was analyzed by the software mentioned above. The four variants identified in intron 2 of *BRCA1* (c.81-3510_81-3505delCTTTTT; c.81-3790G > C; c.81-4100A > C; c.80 + 247C > T) deserve attention. In fact, all four variants identified in this study map in two conserved enhancer regions, that have already been demonstrated to be involved in gene expression regulation (Appendix A) [26]. 

In conclusion, we found only one Pathogenic variant in the *RAD51C* gene, whereas the large majority were classified as VUSs (50), 2 as Likely-benign and 11 as Benign variants. 

The high number of VUSs is due to the fact that very few studies have analyzed these non-coding regions, and for this reason the classification is based only on in silico prediction.

### 2.5. Deep Intronic Regions

The panel of 9071 probes used was designed to optimize coverage of exonic regions, 50 bp flanking each intron-exon junction, 5′UTR, 3′UTR of all the genes of interest and the known regulatory regions of *BRCA1* and *BRCA2*. However, the position of the probes managed to cover additional regions which led us to identify another nine variants in intronic regions (all with a coverage ≥ 50X) that proved to be interesting for the predictions obtained by ‘TFbind’ and ‘Promo’ software and for the information obtained from rVarBase and ENCODE (Appendix A). Two variants are in the intron 61 of *ATM*, one in the intron 16 of *BRIP1*, two in intron 2 respectively of *RAD51C* and *RAD51D* genes and four variants are in the intron 1 of *TP53*. Eight out of nine variants are classified as VUS and four are not registered in dbSNP. 

### 2.6. Demographics and Clinical Features of Patiens with Non-Coding Variants

Non-coding variants were found in 64 patients (excluding two cases of co-occurrence of Pathogenic coding variants). According to this data, in these patients there is a significant increase in both ovarian cancer (χ2-test, *p* < 0.05) and HBOC families (χ2-test, *p* < 0.01), compared to patients with variants in coding regions (Table 1/Table 4).

The mean age of tumor onset for these patients is 44.4. If we compare the coding and non-coding variant carriers in each proband group with the same kind of cancer, the average age of disease onset is completely comparable (Table 1/Table 4). In particular, if we compare the 64 non-coding variant carriers to the 39 Pathogenic coding variant carriers, there is not a statistically significant difference in the proportion of cases with a tumor onset under age 40 (Table 5, χ2-test, *p* > 0.05).

Figure 5 shows the distribution by gene and by gene location of the variants identified respectively in 43 patients with BC, in 12 patients with ovarian cancer, and in 9 cases of patients with multiple tumors (seven with bilateral cancer, one with breast and ovarian cancer, and one with breast and pancreatic cancer). From this data emerges the high frequency of variants identified in the *ATM* gene, as well as in *BRCA1* and *BRCA2*, in patients with BC and the high contribution of variants localized in the 3′UTR of *PTEN* in patients with multiple tumors. In particular, three out of nine variants found in these patients are localized in *PTEN* 3′UTR and the bioinformatics predictions found, for all three, consensus sequences compatible with miRNA binding sites (Figure 5, Appendix A). In patients with ovarian cancer, instead, variants were identified mainly in the *RAD50* and *RAD51C* genes.

Among the 12 cases of TNBC, all with G2 or G3 infiltrating ductal carcinoma, 4 variants out of 19 identified are on the *PTEN* gene: one located at 5′UTR and the others at 3′UTR. Other interesting variants, for the type of feedback we obtained through prediction analysis, are the variants in the 5′UTR of *RAD51D* and *STK11* genes. The complete list of variants found in these patients is shown in Table 6. Four variants are in intronic regions of *BRCA2*. Furthermore, among all the non-coding variants filtered in this work, the only variant in the *RAD51C* gene classified as “Pathogenic” by ACMG criteria was found in one of these patients. 

Twelve patients with multiple variants in non-coding regions were filtered to assess whether the presence of multiple mutations could affect tumor pathogenicity. Ten out of 12 filtered patients have BC and 6 of these have TNBC. According to our data, there is a statistically significant association between TNBC and multiple variants in non-coding regions (χ2-test, *p* < 0.05). In fact, 50% of BC patients with multiple variants in non-coding regions have TNBC, while, among the 52 BC non-coding variants carriers, only 12 (23.1%) have TNBC (χ2-test, *p* < 0.05; Table 7).

## 3. Discussion

This study regards the analysis of coding regions and 50 bp flanking intron–exon junction of 10 BC predisposition genes in a cohort of 450 HBOC patients. For a portion of these patients (120) the screening was extended to additional two genes (*NBN* and *RAD50*), to the 5′UTR and 3′UTR of each gene analyzed and to the known regulatory regions of *BRCA1* and *BRCA2*.Eighty eight coding variants were identified in 91/450 patients (20.2%): 35 of these were classified as pathogenic (class 5) or likely-pathogenic (class 4) and found in 39 patients (8.7%), 37 were classified as VUSs in 41 patients (9.1%) and 16 as “Likely-benign” in 19 patients (4.2%).

*ATM* was the more mutated gene (16 carriers), followed by *CHEK2* (8 carriers), *BRIP1* (7 carriers), *PALB2* (3 carriers). *RAD51C* and *RAD51D* were found mutated in 2 patients/each and *TP53* in one patient only.

Thirteen variants in the intron-exon junctions were identified in 23/450 patients (5.1%) and 51 non-coding variants in 39 out of 120 patients analyzed for this purpose (32.5%).

Our results support what was recently published by the Breast Cancer Association Consortium in a larger and more geographically heterogeneous population consisting of 60,466 women. With a panel of 34 putative susceptibility genes, a significant association between the increased risk of breast cancer and truncating mutations in *ATM*, *CHEK2*, *PALB2*, *BARD1*, *RAD51C*, *RAD51D*, and *TP53* was identified [27]. Similar results were found through a screening of 65,057 white woman with breast cancer where a high or moderately increased risk of disease was found in relation with pathogenic mutations in *ATM*, *CHEK2*, *PALB2*, and *RAD51D* [28].

*ATM*, *CHEK2*, *PALB2*, and *TP53* were confirmed as breast cancer predisposition genes also by the German Consortium for Hereditary Breast and Ovarian Cancer, through a screening of eight cancer predisposition genes conducted on 5589 breast cancer patients negative for pathogenic *BRCA1/2* mutations [29]. An American retrospective study performed on 35,000 women who underwent genetic testing through a 25-gene hereditary cancer panel found the same correlation [30].

The analysis of non-coding regions reports a consistent number of variants never registered previously (18/64 variants). 

In details, in the 5′UTR 2 variants were found, one in *BRCA1* (c.-676t > a) and one in *BRCA2* (c.-1193c > t). Both were registered in BRCA-exchange and LOVD database as VUSs. The in-silico analysis, conducted by ‘TFbind’ and ‘Promo Prediction’, has identified putative consensus sequences for Elk-1, NF-AT1, STAT4 transcription factors, for the first variant, and NRF2, GATA1/2, USF for the second. 

Whereas the 5′UTR and 3′UTR of the *BRCA1/2* genes have been two of the most intensely investigated non–coding regions over the last few years [12,13,31,32,33,34], this is the first fully comprehensive study of 5′and 3′UTRs of so called “minor genes for breast/ovarian cancer susceptibility” and a number of variants were also identified in the 5′UTRs: 4 in the *PTEN* gene and 3 in the *RAD51C* gene and the in-silico predictions obtained regarding their involvement in TF binding are interesting (Appendix A).

In the 3′UTRs of genes other than BRCA1/2, the greatest contribution was due to variants located in the *ATM* (5), *PTEN* (7) and *RAD50* (4) genes. The use of software such as ‘TargetScanHuman’ and ‘miRDB’ has made it possible to identify in those regions binding sites for miRNAs (Appendix A).

The variants localized in splice regions mainly affect the *ATM* (3) gene and, in particular the c.7788+8G >T variant, localized in intron 52 of *ATM*, could modify an exonic ESE site (Appendix A). Another variant of interest is the c.904+5G >T of *RAD51C*, classified by ACMG criteria as ‘Pathogenic’. 

According to our data, variants in the intronic regions have been identified mainly in the *BRCA1* (5), *BRCA2* (7), *ATM* (2), and *RAD50* (2) genes. 

Four out of 5 *BRCA1* variants are localized in intron 2 (c.81-3510_81-3505delCTTTTT; c.81-3790G > C; c.81-4100A > C; c.80+247C > T) and the in-silico predictive analysis led us to identify TF binding sites in those regions (Appendix A). Those variants deserve to be studied in depth through a functional assay because in *BRCA1* intron 2, two conserved enhancer regions (CNS1 and CNS2) are localized and involved in the regulation of gene expression [26].

Eleven out of 12 intronic variants found in *BRCA1/2* genes could be classified as ‘deep intronic’. The deep intronic regions of the BRCA genes are intensely been studied in the recent years. These investigations have led to the identification of variants capable to modify the expression level or the functionality of the BRCA1 protein [35].

Finally, we incidentally found two new variants in deep intronic regions of the *ATM* and *TP53* genes in two patients with TNBC (due to the fact that additional regions that were not of our interest were included in the panel design). The variants are respectively in intron 61 of *ATM* (c.8850+389G > A) and intron 1 of *TP53*(c.-29+475C > A) and are both predicted as TF binding regions. This result underlines how important is to broaden screening to deeper intronic regions (not only in BRCA genes).

By analyzing the characteristics of patients for the type of variants identified, distinguishing between coding variant carriers versus non-coding variant carriers, we can make some interesting considerations.

The average age of breast cancer onset in coding variants carriers is 44 and most of them have breast cancer only family history as shown in Table 1.

A possible association of *ATM-BRIP1-PALB2* and a significant association of *CHEK2* pathogenic/likely-pathogenic coding variants with multiple tumors, including bilateral breast cancer, was observed. These data are in agreement with the findings of German Consortium for Hereditary Breast and Ovarian Cancer, that demonstrated a significant association of *CHEK2* deleterious variants with bilateral breast cancer [29].

Among the 12 cases of TNBC patients, 6 were carriers of pathogenic coding variants: 2 in *CHEK2* and the remaining 4 in the *ATM*, *BRIP1*, *RAD51D* and *TP53* genes respectively (Table 3).

Studies conducted on a larger number of TNBC patients have come to concordant results regarding the association of these cases with pathogenic variants in the *PALB2*, *BRIP1*, *RAD51C*, and *RAD51D* genes [30,36,37,38]. Our results partially agree with these data, having identified pathogenic variants in TNBC patients with *ATM*, *RAD51D*, *BRIP1*, and *TP53* variants. It is likely that the analysis of a higher number of TNBC cases could have led us to a greater degree of concordance. 

However, it is certainly very interesting that 50% of the TNBC cases analyzed had pathogenic coding variants: it is a higher proportion compared to that obtained in other cohorts of TNBC patients filtered in the population solely on the basis of their tumor phenotype [36,37]. It is probable that the higher percentage registered in our cases depends on the double filtering, based not only on patient tumor phenotype but also on the patient breast cancer family history. 

It is known that some TNBCs BRCAWT show the so-called ‘BRCAness’ phenotype, exhibiting clinical and pathological properties similar to BRCA-mutated tumors such as the homologous recombination deficiency and the sensitivity to DNA damaging agents [39,40]. Therefore, our data could be of interest at a therapeutic level. In fact, the data supports what we have already reported in Spugnesi et al. [38] where germline mutations in DNA repair genes, in BRCA1/2 WT tumors, were associated with a group of TNBC patients who responded to anthracyclines/taxanes neoadjuvant therapy. It is becoming increasingly evident that genes involved in DNA repair could become response biomarkers for DNA-damaging therapy in these patients [41].

Although we cannot make any consideration about the variants identified in ovarian cancer, having only analyzed four patients with ovarian cancer and two with breast and ovarian cancer, the detection of pathogenic variants in *RAD51C* and *ATM* is in agreement with what is reported in a case-control study performed on 2051 women with ovarian cancer. The authors of this paper identified an increased risk for ovarian cancer associated with *MSH6*, *RAD51C*, *TP53*, and *ATM* [42].

Analyzing the characteristics of non-coding variants carriers, we recorded an average age of disease onset of 44.4 absolutely similar to the average age of disease onset in coding variant carriers for each proband’s group with the same cancer type. Furthermore, there is not a statistically significant difference in the proportion of cases with a tumor onset under the age of 40 between the two groups (χ2-test, *p* > 0.05; Table 5).

A statistically significant increase in cases of ovarian cancer and of HBOC families in non-coding variant carriers compared to coding variant carriers was found (Table 1/Table 4). This is an interesting observation but very preliminary considering the limited number of samples analyzed, and it needs to be further investigated.

One of the results that emerges clearly when analyzing the characteristics of non-coding variant carriers, is the relationship between variants in 3′UTR of *PTEN* and multiple tumors or TNBC tumor phenotype. This result is in agreement with what has already been described in previous studies in which the deregulation of *PTEN* expression, via binding sites for RNA-binding-protein and miRNA located in 3′UTR, was associated with aggressive phenotype and poor outcome for BC patients [16,17]. Moreover, four out of seven *BRCA2* deep intronic variants identified (c.9257-3610G > A, c.681+697C > T, c.317-1021A > G, c.317-512A > T) have been found in TNBC patients.

Finally, by analyzing the 12 patients carrying multiple non-coding variants we observed a statistically significant association with a TNBC phenotype (χ2-test, *p* < 0.05).

This work has the merit of having highlighted certain variants never recorded in the coding and regulatory regions which are worthy of further study. It also underlines the importance, increasingly evident in recent years, of also screening the regulatory regions in patients with clearly familial tumors with no relevant variation in the coding sequences of the main breast/ovarian cancer predisposition genes. Certainly, in the future it will be necessary to perform a segregation analysis in order to understand the real involvement of certain genomic regions in breast and ovarian cancer predisposition. This work surely could help to pave the way for other in-depth projects on the functional meaning of the variants identified, by providing, thanks to the many notes obtained, a lot of useful information for a better understanding of their clinical significance.

## 4. Materials and Methods

### 4.1. Selection of Patients Included in the Study

We selected a total of 450 patients eligible for this study who underwent genetic counseling between 2001 and 2019 at five different Italian centers distributed as follows:-163 patients from University Hospital of Pisa-210 patients from Azienda Tutela Salute Sardegna, ASST of Cagliari-55 patients from Ospedale di Circolo, ASST-Sette Laghi of Varese-13 patients from University Hospital of Modena-9 patients from IRCCS Istituto Tumori “Giovanni Paolo II” of Bari

All patients signed an informed consent. The eligibility criteria included:-family history of BC in at least two generations: (a) two or more BC cases in first degree relatives, one of which diagnosed before age 40 (b) one ovarian cancer case associated with a BC in a first degree relative at any age;-a priori risk of being carriers of BRCA1/2 mutation >10% calculated with BRCAPRO (http//www4.utsouthwestern.edu/breast-health/cagene, accessed on March 2017–December 2019 ) and BOADICEA (https://pluto.srl.cam.ac.uk/cgi-bin/BD2/v2/bd.cgi, accessed on March 2017–December 2019) software;-to be wild type for point mutations and large rearrangements in the BRCA1/2 coding regions.

### 4.2. Library Preparation and NGS Sequencing

DNA was extracted from blood samples. Genomic DNA screening was carried out with a NGS target resequencing approach on the Illumina MiSeq machine. 450 patients were analyzed with a custom panel optimized to sequence the coding regions and 50 bp flanking each intron-exon junction of 10 genes involved in breast and ovarian cancer predisposition: *ATM* (NM_000051.3), BRIP1 (NM_032043.2), *CDH1* (NM_004360.3), *CHECK2* (NM_007194.3), *PALB2* (NM_024675.3), *PTEN* (NM_000314.6), *RAD51C* (NM_058216.2), *RAD51D* (NM_002878.3), *STK11* (NM_000455.4), *TP53* (NM_000546.5). For a subset of 120 patients (43 from Pisa, 55 from Varese, 13 from Modena and 9 from Bari), the screening was also extended to *NBN* (NM_002485.5) and *RAD50* (NM_005732) genes, including 5′UTR and 3′UTR of each gene and the known regulatory regions of *BRCA1* (NM_007294.4) and *BRCA2* (NM_000059) [12,13,14,15,26]. To amplify the target sequences, a panel of 9071 probes was designed using SureSelect DNA Advanced Design Wizard in order to ensure the total coverage of 178.155 kb corresponding to 99% of the entire region of interest (Appendix A). Library preparation was performed according to SureSelectQXT Target Enrichment for Illumina Multiplexed Sequencing protocol (SureSelectQXT Reagent Kit, Agilent Technologies, Santa Clara, CA, USA). To assess library DNA quantity and quality after fragmentation and adaptor-tag addiction phase and after amplification of the indexed library, we used the Agilent 2100 Bioanalyzer. Following purification and quantification of pooled libraries, we sequenced them by 2 × 100 bp paired-end reads on a Miseq (Illumina, San Diego, CA, USA) instrument. Sequence reads allocated to each individual were aligned to the human reference sequence (hg19) using Agilent Sure Call (ver.4.1.1). All the regions considered for the analysis had a coverage ≥50X and a Quality Score (QS) ≥30. Sanger sequencing was performed to confirm the identified variants in lymphocyte DNA.

### 4.3. Bioinformatic Annotation and Prioritization of the Identified Variants

Each identified variant was evaluated in terms of Minor Allele Frequency (MAF): reference allelic frequency is that of total population reported in 1000 Genomes (http://www.internationalgenome.org, accessed on January 2020) and in gnomAD (https://gnomad.broadinstitute.org, accessed on January 2020). All variants with a MAF >1% were excluded from further analysis as they were considered common polymorphisms. To speed up the reading of the variants and their filtering on the basis of the known annotations, we used BaseSpace Variant Interpreter platform (https://variantinterpreter.informatics.illumina.com, accessed on January 2020). All variants were also evaluated in term of evolutionary conservation profile (GERP score) [43]. The clinical significance of each variant was annotated considering information from ClinVar (https://www.ncbi.nlm.nih.gov/clinvar, accessed on August 2020) [44] and according to ACMG (American College of Medical Genetics) criteria [19,45] also used by Varsome (https://varsome.com, accessed on August 2020) [46]. The latter helped us mostly to summarize all the known information and the pathogenicity predictions of the identified variants. 

For the coding variant, we annotated the possible impact of an amino acid substitution on the protein structure and function obtained by PolyPhen2 (http://genetics.bwh.harvard.edu/pph2, accessed on May 2020) and PROVEAN (http://provean.jcvi.org/index.php, accessed on May 2020) software [47,48]. For the non-coding variants, in order to obtain information about chromatin states of the surrounding regions, methylation and acetylation status and hypersensitivity to DNAseI, we used rVarBase (http://rv.psych.ac.cn, accessed on February 2020) and ENCODE (https://www.encodeproject.org, accessed on February 2020) database [49,50,51].

To predict possible TF binding site in 5′UTR and in intronic enhancer regions, we emploied three different software: Promo (http://alggen.lsi.upc.es/cgi-bin/promo_v3/promo/promoinit.cgi?dirDB=TF_8.3, accessed on February 2020) [52], Tfbind (http://tfbind.hgc.jp/) and Lasagna-Search2.0 (https://biogrid-lasagna.engr.uconn.edu/lasagna_search/index.php, accessed on February 2020) [53]. The presence of miRNA binding site in 3′UTR variants, was evaluated by TargetScanHuman (http://www.targetscan.org/vert_72, accessed on February 2020) and miRDB (http://www.mirdb.org, accessed on February 2020) [54,55]. Finally, in order to identify consensus regions for potential splice sites, we made use of Human Splicing Finder (http://www.umd.be/HSF, accessed on February 2020) [56]. All the programs used to obtain predictions about the variants found are free software and they are summarized in Table 8.

All this information allowed us to have more complete annotations of the variants, making it possible to suggest future functional tests, especially those never reported in literature or in databases such as ENIGMA (https://enigmaconsortium.org, accessed on December 2020), BRCA Exchange (https://brcaexchange.org, accessed on December 2020), LOVD (https://www.lovd.nl, accessed on December 2020), MgeND (https://mgend.med.kyoto-u.ac.jp, accessed on December 2020), and ClinVar Miner (https://clinvarminer.genetics.utah.edu, accessed on December 2020). For each variant, the dbSNP (https://www.ncbi.nlm.nih.gov/snp, accessed on January 2021) encoding was also checked.

### 4.4. Statistical Analysis 

Tumor type percentages, family distribution, and age at diagnosis in coding and non-coding variant carriers were compared by χ2-test. *p* < 0.05% was considered statistically significant. The same test was used to estimate the effect of Pathogenic, VUS, and Likely Benign variants on clinical characteristics of coding variant carriers and to evaluate the presence of multiple variants in more aggressive tumors in non-coding variant carriers. 

The gene association with tumor aggressiveness was performed through the Fisher’s test. The results were considered significant with a threshold of *p* < 0.05%. All analyses were performed with the free software Social Science Statistics (https://www.socscistatistics.com, accessed on January 2021).

## Figures and Tables

**Figure 1 ijms-22-07693-f001:**
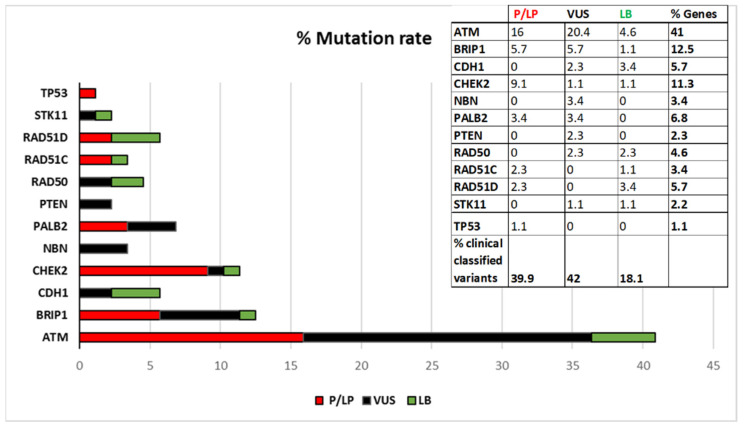
Percentage of identified coding variants distributed by gene and by clinical classification. P/LP = pathogenic/likely pathogenic, VUS = variant of uncertain significance, LB = likely benign.

**Figure 2 ijms-22-07693-f002:**
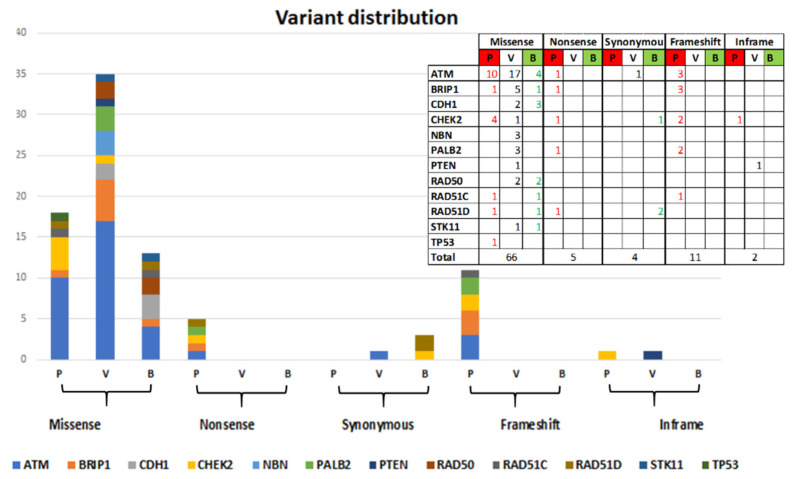
Distribution of 88 variants by effect and by clinical classification (P = Pathogenic/Likely-pathogenic, V = VUS, B = Likely-benign).

**Figure 3 ijms-22-07693-f003:**
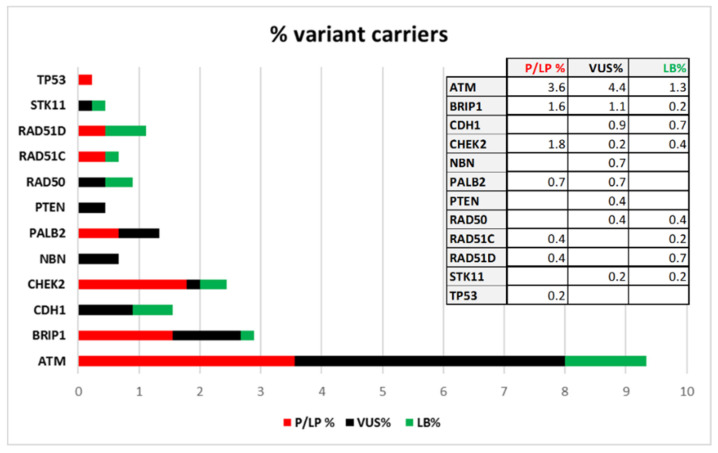
Percentages of coding-variant carriers distributed by gene and by clinical classification out of the total of 450 cases screened.

**Figure 4 ijms-22-07693-f004:**
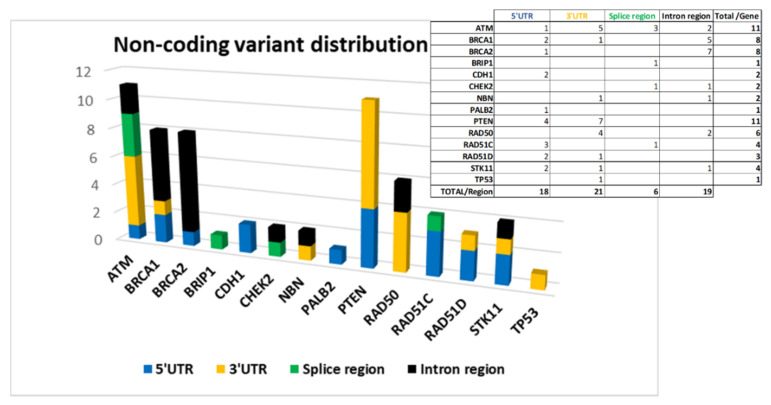
Non-coding variant distribution by gene and by gene region (5′UTR, 3′UTR, Splice regions, and Intron regions).

**Figure 5 ijms-22-07693-f005:**
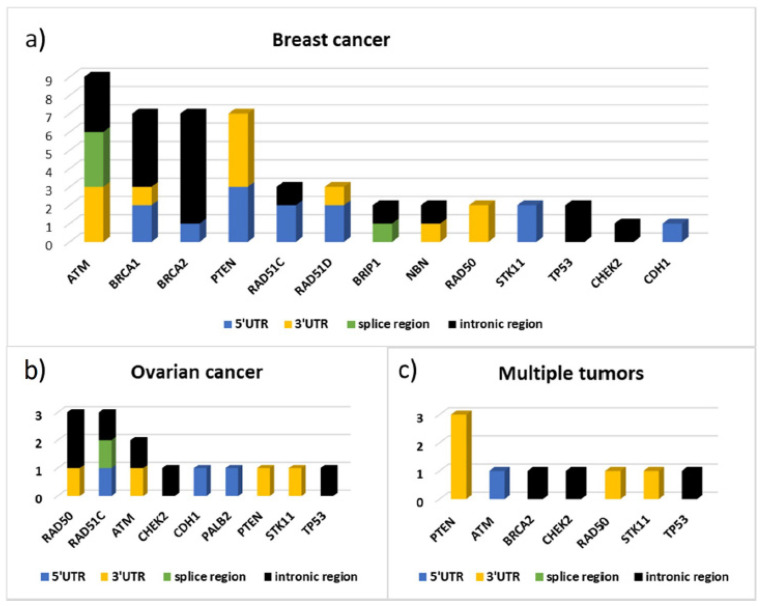
Variant distribution by gene and by gene region (5′UTR, 3′UTR, Splice regions and Intron regions) respectively in patients with breast cancer (**a**), ovarian cancer (**b**), and with multiple tumors (**c**).

**Table 1 ijms-22-07693-t001:** Coding variant carrier characteristics: percentage distribution of probands according to the cancer they are affected by and to their cancer family history. The number of patients is shown in brackets. The average age at disease onset is reported for each proband kind of cancer (for patients with bilateral breast cancer and multiple tumors, the age of onset of initial tumor has been considered).

Coding Variant Carrier Characteristics		
Patient number	91	
Average age of cancer onset	44	
**Proband cancer kind**		**Average age at onset**
Breast cancer	82.4% (75)	42.8
Bilateral Breast Cancer	8.8% (8)	45.7
Ovarian cancer	4.4% (4)	47.7
Breast and ovarian cancer	2.2% (2)	49.5
Breast/Ovarian Cancer and other tumors	2.2% (2)	54.5
**Proband cancer history**		
HBC	48.3% (44)	
HBOC	12.1% (11)	
EOBC	18.7% (17)	
Other HBCS	20.9% (19)	

**Table 2 ijms-22-07693-t002:** List of variants found among the 12 cases of patients with multiple tumors in coding-variant carriers. The table reports the protein change of variant, the clinical classification (P = Pathogenic, LP = Likely-pathogenic, VUS = Variant of Unknown Clinical Significance, and LB = Likely-benign), the variant frequency among these 12 cases, the patient cancer type in which they were detected and the eventual database registration. The variants are reported in HGVS nomenclature.

Variants in 12 Cases of Multiple Tumors							
Variant	Protein Change	Gene	Classification	N.patients	Tumor	dbSNP	Database
c.3894dupT	p.Ala1299CysfsTer3	ATM	P	1	Bil. Breast	rs587781823	ClinVar Miner
c.8368delA	p.Arg2790AspfsTer16	ATM	P	1	Bil. Breast		
c.3284G > A	p.Arg1095Lys	ATM	LP	1	Br/Ov	rs587781815	LOVD
c.2237_2240delTCAA	p.Ile746AsnfsTer12	BRIP1	P	1	Bil.Breast	rs587782726	MGeND
c.1312G > T	p.Val438Phe	CHEK2	LP	1	Bil.Breast	rs58778017	LOVD
c.483_485delAGA	p.Glu204del	CHEK2	LP	1	Bil.Breast/Pancr.	rs587782008	MGeND
c.507delT	p.Asp169GlufsTer5	CHEK2	LP	1	Br/Ov		
c.1430delC	p.Thr477LysfsTer7	CHEK2	LP	1	Bil.Breast		
c.1451T > A	p.Leu484Ter	PALB2	P	1	Bil.Breast	rs786203714	ClinVar Miner
c.1516G > T	p.Gly506Cys	ATM	VUS	1	Bil.Breast	rs587779816	LOVD
c.131G > T	p.Arg44Leu	CDH1	LB	1	Bil.Breast	rs1375178645	
c.132C > T	p.Ser44=	CHEK2	LB	1	Bil.Breast	rs199715101	ClinVar Miner

**Table 3 ijms-22-07693-t003:** List of variants identified in 12 cases of patients with Triple-Negative Breast cancer (TNBC). The table also reports the protein change for each variant, their clinical classification (P = Pathogenic, LP = Likely-pathogenic, VUS = Variant of Unknown clinical Significance, LB = Likely-benign), their frequency identified in these patients and whether they have been previously recorded in the database LOVD. The variants are reported in HGVS nomenclature.

Variants in 12 Cases of TNBC						
Variant	Protein Change	Gene	Classification	N.patients	dbSNP	Database
c.6067G > A	p.Gly2023Arg	ATM	P	1	rs11212587	LOVD
c.2392C > T	p.Arg798Ter	BRIP1	P	1	rs137852986	LOVD
c.409C > T	p.Arg137Ter	CHEK2	P	1	rs730881701	LOVD
c.1118A > G	p.Gln373Arg	CHEK2	LP	1	rs894075046	LOVD
c.694C > T	p.Arg232Ter	RAD51D	P	1	rs587780104	LOVD
c.817C > G	p.Arg273Gly	TP53	P	1	rs121913343	LOVD
c.1272T > C	p.Pro424=	ATM	VUS	1	rs35578748	LOVD
c.4703A > G	p.His1568Arg	ATM	VUS	1	rs368830730	LOVD
c.4060C > A	p.Pro1354Thr	ATM	VUS	1	rs145119475	LOVD
c.17_18insGCG	p.Lys6_Glu7insArg	PTEN	VUS	1		
c.2119T > C	p.Ser707Pro	ATM	LB	1	rs4986761	LOVD
c.1810C > T	p.Pro604Ser	ATM	LB	1	rs2227922	LOVD

**Table 4 ijms-22-07693-t004:** Non-coding variant carrier characteristics: percentage distribution of probands according to cancer type and cancer family history. The number of patients is shown in brackets. The average age at disease onset is reported for each proband cancer type (for patients with bilateral breast cancer and multiple tumors, the age of onset of initial tumor has been considered).

Non-Coding Variant Carrier Characteristics		
Patient number	64	
Average age of cancer onset	44.4	
**Proband cancer kind**		**Average age at onset**
Breast cancer	67.2% (43)	43.9
Bilateral Breast Cancer	10.9% (7)	43.1
Ovarian cancer	18.7% (12)	44
Breast and ovarian cancer	1.6% (1)	48
Breast/Ovarian Cancer and other tumors	1.6% (1)	49
**Proband cancer history**		
HBC	53.1% (34)	
HBOC	29.7% (19)	
EOBC	4.7% (3)	
Other HBCS	12.5% (8)	

**Table 5 ijms-22-07693-t005:** Age of tumor onset: comparison between Pathogenic coding variant carriers and non-coding variant carriers.

	Pathogenic Coding Variant Carriers	Non-Coding Variant Carriers
≤40	15	22
>40	24	42
Total	39	64

**Table 6 ijms-22-07693-t006:** Variants identified in the 12 patients with TNBC among the non-coding-variant carriers. The table shows the variant position in the gene (ss = splice site), the clinical classification (P = Pathogenic, VUS = Variant of Unknown clinical Significance, B = Benign), the frequency of each variant and any database registration. The variants are reported in HGVS nomenclature and those highlighted with the same color belong to the same patient.

Variants in 12 Cases of TNBC						
Variant	Gene	Position	Classification	Patient	dbSNP	Database
c.-323C > T	PTEN	5’UTR	VUS	Varese 36		
c.-116C > T	RAD51D	5’UTR	VUS	Varese 36		
c.*1122_*1123insT	PTEN	3’UTR	VUS	Varese 36		
c.9257-3610G > A	BRCA2	Intron 24	VUS	Varese 36	rs17077519|	BRCAexchange
c.-239C > T	RAD51D	5’UTR	LB	Varese 55	rs550346401	
c.681+697C > T	BRCA2	Intron 8	VUS	Varese 55	rs11571632	BRCAexchange
c.-636C > T	STK11	5’UTR	VUS	Pisa 1489	rs981125953	
c.*1674 > G	RAD50	3’UTR	VUS	Pisa 1489		
c.*26A > G	ATM	3’UTR	VUS	Pisa 2348	rs1247328981	
c.*724dupT	NBN	3’UTR	VUS	Varese 10	rs564878448	
c.*4606_*4609delGCTT	PTEN	3’UTR	B	Bari 1		
c.*1152C > T	PTEN	3’UTR	B	Varese 17	rs552354954	
c.7788+8G > T	ATM	Intron 52ss	VUS	Bari 4	rs112775908	LOVD
c.317-1021A > G	BRCA2	Intron 3	VUS	Bari 4	rs11571604	BRCAexchange
c.8850+389G > A	ATM	Intron 61	LB	Varese 14		
c.317-512A > T	BRCA2	Intron 3	VUS	Varese 14		
c.320-5T > C	CHEK2	Intron 24ss	VUS	Pisa 2280		LOVD
c.904+5G > T	RAD51C	Intron 6ss	LP	Pisa 1564	rs587782702	LOVD
c.-29+475C > A	TP53	Intron 1	VUS	Varese 20		

**Table 7 ijms-22-07693-t007:** Patients with multiple non-coding variants, age at disease diagnosis and tumor characteristics. The table also reports the cases of co-occurrence of VUS and Likely-benign coding variants with multiple non-coding variants in the same patient. The variants are reported in HGVS nomenclature.

Patients	Non-Coding Variants	Coding Variants	Cancer	Age at Diagnosis	Histological Characteristics	TNBC
Pisa 710	c.-1193C > T BRCA2; c.-408T > A BRCA1		Breast	57	infiltrating ductal carcinoma	
Pisa 1489	c.*1674A > G RAD50; c.-636C > T STK11		Breast	50	infiltrating ductal carcinoma	V
Bari 4	c.7788+8G > T ATM, c.317-1021A > G BRCA1	c.2074G > A CDH1, c.17_18insGCG PTEN	Breast	37	infiltrating ductal carcinoma	V
Varese 7	c.-138G > C PALB2, c.-29+622G > C TP53	c.1094G > A RAD50	Ovary	48	serous carcinoma	
Varese 8	c.81-4100A > C BRCA1, c.*114A > G RAD50	c.4060C > A ATM	Breast	45	infiltrating ductal carcinoma	
Varese 14	c.8850+389G > A ATM, c.317-512A > T BRCA2		Breast	41	infiltrating ductal carcinoma	V
Varese 36	c.9257-3610G>A BRCA2, c.-323C > T PTEN, c.*1122_*1123insT PTEN, c.-116C > T RAD51D		Breast	44	infiltrating ductal carcinoma	V
Varese 43	c.560+19_560+22delTACT ATM, c.904+5G > T RAD51C		Ovary	49	serous carcinoma	
Varese 45	c.*2842A > G ATM, c.*750A > G BRCA1, c.*106G > A RAD51D	c.1810C > T ATM, c.6226A > G ATM	Breast	46	infiltrating ductal carcinoma	
Varese 51	c.8850+689A > C ATM, c.*537A > C ATM		Breast	34	infiltrating ductal carcinoma	
Varese 55	c.681+697C > T BRCA2,c.-239C > T RAD51D	c.1810C > T ATM	Breast	45	infiltrating ductal carcinoma	V
Pisa 2582	c.-24A>C ATM, c.*261C>T STK11		Bil.Breast	46	infiltrating ductal carcinoma	

**Table 8 ijms-22-07693-t008:** Tools used for variant analysis, annotations and prioritization.

Coding Variant Annotations	Non-Coding Variant Annotations
Variant information from human genomic data (Variant Interpreter, Varsome)	Variant information from human genomic data (Variant Interpreter, Varsome)
Relationships among human variants and phenotypes, clinical significance (ClinVar, ACMG criteria-Varsome)	Regulatory annotations, chromatin states of the surrounding regions, methylation and acetylation status, hypersensitivity to the DNAseI (rVarBase, Encode)
Possible impact of an amino acid substitution on the structure and function of a human protein (PholyPhen2, PROVEAN)	TF binding sites ( PromoPrediction, TF Bind, Lasagna)
	miRNA binding sites (TargetScan Human, miRDB)
	Consensus regions for potential splice sites (Human Splicing Finder)

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
