# Peer review of "Detection of Germline Variants in 450 Breast/Ovarian Cancer Families with a Multi-Gene Panel Including Coding and Regulatory Regions"

_ijms, 2021, doi:10.3390/ijms22147693_

Round 1
Reviewer 1 Report
The authors present a study of the prevalence of mutations in 12 moderate penetrance genes for breast and / or ovarian cancer. From the reading of the manuscript, (page 3 lines 123 to 127) a mutational screening was performed for 450 patients in the coding regions and flanking regions of 50 bp of 10 genes (ATM, BRIP1, CDH1, CHECK2, PALB2, PTEN, 124 RAD51C, RAD51D, STK11, TP53), and only in 120 patients have they been examined for RAD50 and NBS and for the regulatory regions (5'UTR and 3'UTR) of the 12 genes plus BRCA1 and BRCA2.
The results describe that the analysis was performed on 450 patients, but only 120 of them had 12 fully analyzed genes. This issue is clearly described in introduction (page 3 lines 111 to 120) and in material and methods, but neither in the results nor in the discussion (page 11 lines 359 to 360). It must be addressed correctly in all sections.
Other issues to correct are:
Page 7 line 243 “…The screening of non-coding regions has allowed us to identify 64 variants…” whereas in line 246 are 66 variants. One of the two figures is incorrect.
Page 12 lines 418-419. “Finally, we incidentally (due to panel design additional regions not of our interest were included) found 2 new variants in deep intronic regions of the ATM and TP53 genes in two patients with TNBC.” It is grammatically incorrect. It sounds better “Finally, we incidentally found 2 new variants in deep intronic regions of the ATM and TP53 genes in two patients with TNBC (due to the fact that additional regions that were not of our interest were included in the panel design).
Author Response
- Dear Reviewer as you suggested we try to improve the description of patients analyzed in the Discussion by specifying, as in the previous sections, that the analysis was carried out on 10 BC predisposition genes in a cohort of 450 HBOC patients whereas the full screening was performed in 120 only of those patients (lines 372-373 latest version of the manuscript)
- In the line 257 (246 in the previous version) at page 7 there was a typo: the non-coding variants identified were 64 as reported in figure 4.
- Regarding the lines 437-438 at page 13 (418-419 at page 12 in the previous version), we agree with you and we accept the correction.
Reviewer 2 Report
General comment: It is an interesting article. For easy interpretation of potentially statistically significance findings, the authors should better describe the study objectives, hypotheses, and variables measured to test the hypotheses in the introduction and method section. Also, please add a supplemental table with a list of all variants, a detailed classification (IARC vs ACMG) and whether the variants are present in ClinVar.

Author Response
- Dear Reviewer we thank you to help us to increase the value of our study by suggesting to add, at the end of the introduction, a few lines to better specify the purpose of the study, our hypotheses and the variables measured to verify the hypotheses . The specific tests used are reported in the paragraph “Statistical analysis” in Materials and Methods section.
- Regarding the supplemental table with the list of all variants, we already included in the “Supplementary Materials” two tables with a complete list of all coding and non-coding variants identified. The tables report for each variant the ACMG, IARC and other database classification and whether the variant is present in dbSNPs. Moreover we predict the pathogenicity of each variant by using different specific software according to variant position.